# A Model-Driven Channel Estimation Method for Millimeter-Wave Massive MIMO Systems

**DOI:** 10.3390/s23052638

**Published:** 2023-02-27

**Authors:** Qingli Liu, Yangyang Li, Jiaxu Sun

**Affiliations:** Communication and Network Laboratory, Dalian University, Dalian 116622, China

**Keywords:** millimeter wave, massive MIMO, channel estimation, beam squint, model driven

## Abstract

Aiming at the problem of low estimation accuracy under a low signal-to-noise ratio due to the failure to consider the “beam squint” effect in millimeter-wave broadband systems, this paper proposes a model-driven channel estimation method for millimeter-wave massive MIMO broadband systems. This method considers the “beam squint” effect and applies the iterative shrinkage threshold algorithm to the deep iterative network. First, the millimeter-wave channel matrix is transformed into a transform domain with sparse features through training data learning to obtain a sparse matrix. Secondly, a contraction threshold network based on an attention mechanism is proposed in the phase of beam domain denoising. The network selects a set of optimal thresholds according to feature adaptation, which can be applied to different signal-to-noise ratios to achieve a better denoising effect. Finally, the residual network and the shrinkage threshold network are jointly optimized to accelerate the convergence speed of the network. The simulation results show that the convergence speed is increased by 10% and the channel estimation accuracy is increased by 17.28% on average under different signal-to-noise ratios.

## 1. Introduction

At present, the fifth-generation (5G) wireless communication network technology has entered a critical period of large-scale application, and the ultimate goal is to realize the mobile Internet and the Internet of Everything [1]. However, the 5G communication system cannot meet all the needs of future wireless communication systems, especially the explosive growth of mobile data traffic and the need for business diversification. Many universities and some foreign scholars have started the research and development of the sixth-generation (6G) wireless communication network technology [2]. To meet the application requirements of the ultra-high transmission rate (huge traffic) and ultra-high connection density (giant connection), the spectrum needs all spectrums below 6 GHz, i.e., millimeter wave, terahertz, and optical band. While the spectrum resources are abundant, its path loss is also high. To overcome the huge attenuation in the millimeter-wave transmission path, the millimeter-wave technology is usually used in combination with massive multiple-input multiple-output (MIMO) technology. Higher energy efficiency and spectrum efficiency can be provided by using large-scale antennas [3]. In recent years, a new digital modulation technique, spatial modulation, has also been extensively studied, which strikes a good balance between spectral efficiency and energy efficiency [4,5]. At the same time, the channel estimation of this single input stream multiple input multiple output technology (SM-MIMO) has also been studied [6]. Therefore, accurate channel state information is indispensable for any communication system, especially for millimeter-wave MIMO-OFDM broadband systems with “beam squint” effect; accurate CSI is the main factor affecting the performance of the system, so accurate channel estimation becomes a key problem to be solved.

Aiming at the problem of channel estimation in the millimeter-wave massive MIMO system, relevant scholars have used the sparse characteristic of millimeter-wave propagation [7] to adopt the restoration method of compressed sensing, implementing algorithms such as Orthogonal Matching Pursuit (OMP) [8,9,10], Basis Pursuit De-Noising (BPDN) [11], and some other similar algorithms [12]. However, the performance of the compressed sensing algorithm is greatly affected by the prior sparsity and the iteration step size, and the anti-noise ability of the algorithm is poor in the environment of low signal-to-noise ratio, so it is not suitable for the channel estimation of the millimeter-wave massive MIMO broadband system. The authors of [13] proposed a Support Detection (SD) method by using the structural characteristics of the beam space channel of a millimeter wave. This method decomposes the channel estimation problem into a series of sub-problems, reducing pilot overhead and ensuring reliable performance. In [14], a Cosparse Analysis Approximate Message Passing (SCAMPI) algorithm based on isomorphism analysis was proposed. This method treats the channel matrix as a two-dimensional image, transforms the channel estimation problem into an image reconstruction problem, and improves channel estimation accuracy. However, the methods proposed in the above documents are all applicable to narrowband systems, but in practice, to achieve higher data rates, millimeter-wave systems are more likely to be broadband. For broadband systems, [15] proposed a method called Synchronous Orthogonal Matching Pursuit (SOMP), which first estimates the support of the wideband beam spatial channel for some subcarriers, then creates public support, and finally restores the entire beam space channel. Some other similar algorithms [16,17] also assume common support, ignoring the “beam squint” effect [18] in wideband millimeter-wave systems; i.e., larger bandwidths cause subcarrier directions to change less at different frequencies. Large bandwidths in turn lead to differences in spatial channels, so the assumed common support is wrong. The authors of [19] proved that each path component in a broadband system has a unique frequency-dependent sparse structure, and proposed a Continuous Support Detection (SSD) method based on applying continuous interference cancellation. This method estimates all sparse path components one by one, but because only the least squares algorithm is used to recover the channel vector, the estimation accuracy in the low SNR area is low. Due to the huge path loss and limited transmit power, the uplink Signal–Noise Ratio (SNR) is usually low [20]. In [21], the authors proposed a model-driven unsupervised generalized expectation consensus (Learned Denoising-based GEC, LDGEC) signal recovery network based on learning denoising for broadband beam space channel estimation, which combines the denoising convolutional neural network DnCNN into the generalized expectation consistent signal recovery algorithm, although it has achieved better performance than the algorithm based on compressed sensing, it can only obtain −7 dB NMSE performance when SNR = 0, so there is still room for improvement in the estimation accuracy under low SNR.

In view of the above method, the channel estimation accuracy is low under a low signal-to-noise ratio for wideband systems with “beam squint” effect, and this paper proposes a new channel estimation method based on Adaptive Threshold Learning Iterative Shrinkage Threshold Network (ALISTA-Net). The method uses model-driven deep learning, based on the iterative shrinkage threshold algorithm, and the algorithm is expanded into the deep iterative network, and the millimeter-wave channel matrix is transformed into the sparse transform domain through training data learning. In the beam domain denoising stage, the Shrinkage Threshold Network (ST-Net) based on the attention mechanism is proposed. The network learns from the idea of the attention mechanism to provide a set of thresholds adaptively through feature learning so that the model is suitable for situations with different noise contents. The denoising effect under different signal-to-noise ratios is used to improve the accuracy of channel estimation; at the same time, to improve the network convergence speed, residual learning is introduced into the network structure. Simulation experiments show that compared with the existing channel estimation algorithms for broadband systems, this method has achieved higher estimation accuracy under different SNRs.

The structure of this paper is divided into the following parts: Section 2 introduces the system model and the description of channel estimation. Section 3 introduces the structure of the ALISTA-Net network proposed in this paper. Section 4 analyzes and discusses the experimental results. Section 5 introduces the conclusion of this paper.

## 2. System Model and Problem Description

### 2.1. System Model

In this paper, in Time Division Duplexing (TDD) mode, a wide-band beam space massive MIMO-OFDM system based on lens array is considered, as shown in Figure 1. The signals are received by the antenna at the receiving end. These signals are processed by radio frequency to obtain baseband analog signals, and then the analog signals are converted into digital signals and then processed synchronously. After the cyclic prefix is removed, they are demodulated by FFT and then processed by baseband signals.

The system is equipped with an N element-lens antenna array and a base station with RF chains to simultaneously serve single-antenna users. This paper considers the widely used Saleh–Valenzuela channel model, then the N×1 spatial channel of the user at the subcarrier m(m=1,2,…,M) can be expressed as
(1)$$h_{m}=\sqrt{\frac{N}{L}}\sum_{l=1}^{L}\beta_{l}e^{-j2\pi \tau_{l}f_{m}}\alpha (\phi_{l,m})$$

L is the number of paths, and $\beta_{l}$ and $\tau_{l}$ are the complex gain and time delay of the l-th path, respectively. Furthermore, $\alpha (\phi_{l,m})$ is the array response vector, and $\phi_{l,m}$ is the spatial direction at the subcarrier m, defined as
(2)$$\phi_{l,m}=\frac{f_{m}}{c}d\sin\theta_{l}$$

Among them, $f_{m}=f_{c}+\frac{f_{b}}{M}+\left(m-1-\frac{M-1}{2}\right)$ is the frequency of the subcarrier m, $f_{c}$ and $f_{b}$ are the carrier frequency and bandwidth, respectively, c is the speed of light, $\theta_{l}$ is the physical direction, and $d=0.5\cdot c/f_{c}$ is the antenna spacing. In millimeter-wave broadband systems, a larger $f_{b}$ will lead to $\phi_{l,m}$ a larger change in subcarriers of different frequencies, which in turn will lead to a difference in the spatial channel $h_{m}$, which is called the “beam squint” effect. When the base station is equipped with a uniform linear lens antenna array, the array response vector $\alpha (\phi_{l,m})$ can be expressed as
(3)$$\alpha \left(\phi_{l,m}\right)=e^{-j2\pi \phi_{l,m}P_{a}}$$

$P_{a}=[-\frac{N-1}{2},-\frac{N+1}{2},\ldots ,\frac{N-1}{2}]$ denotes the indices of different antennas. The lens antenna array shown in Figure 1 can transform the space domain in (1) into the beam space domain. The lens antenna array is similar to the N×N spatial discrete Fourier transform (DFT) matrix F. Therefore, the wideband beam spatial channel ${\tilde{h}}_{m}$ at the subcarrier m is expressed as
(4)$${\tilde{h}}_{m}=F^{H}h_{m}=\sqrt{\frac{N}{L}}\sum_{l=1}^{L}\beta_{l}e^{-j2\pi \tau_{l}f_{m}}{\tilde{c}}_{l,m}$$
where ${\tilde{c}}_{l,m}$ denotes the l-th path component at the subcarrier m in the beam space, and ${\tilde{c}}_{l,m}$ is determined $\phi_{l,m}$ by
(5)$${\tilde{c}}_{l,m}=F^{H}\alpha \left(\phi_{l,m}\right)={\left[\Xi \left(\phi_{l,m}-{\bar{\phi }}_{1}\right),\Xi \left(\phi_{l,m}-{\bar{\phi }}_{2}\right),\ldots ,\Xi \left(\phi_{l,m}-{\bar{\phi }}_{N}\right)\right]}^{⊤}$$

$\Xi (x)=\frac{\sin N\pi x}{\sin\pi x}$ is the function of Dirichlet, and ${\bar{\phi }}_{n}=\frac{1}{N}\left(n-\frac{N+1}{2}\right)(n=1,2,\ldots ,N)$ is the predefined spatial orientation of the lens antenna array.

### 2.2. Problem Description

The received signal vector at the base station can be expressed as
(6)$$y_{m,q}=W_{q}({\tilde{h}}_{m}s_{m,q}+n_{m,q})$$

Among them, ym,q∈ℂ NRF×1 and sm,q is defined as the transmission pilot of the subcarrier m at the moment q(q=1,2,…,Q) (each user only sends one pilot at each moment). nm,q∼NC0,σ2I denotes a Gaussian noise vector, σ2 denoting the noise power. Wq∈ℂ NRF×N is a fixed adaptive selection network for different subcarriers. Since the pilot signal is known at the receiver side, it is assumed that sm,q=1. After Q instants of pilot transmission, the total measurement signal can be obtained as
(7)y¯m=ym,1⊤,…,ym,Q⊤⊤=W¯h˜m+nmeq

W¯=W1⊤,W2⊤,…,WQ⊤⊤∈ℂ QNRF×N is expressed as the overall combination matrix, and nmeq=W1nm,1⊤,…,WQnm,Q⊤⊤ is expressed as the effective noise vector. In this paper, a low-cost 1-bit phase shifter is used in the adaptive selection network, where elements of W¯ are randomly selected in the set 1QNRF{−1,+1}. By stacking the M beam space channel vectors into a matrix and converting the complex values to real values, the following signal recovery problem description can be obtained
(8)Y=W¯H+N

Y represents the received signal, and H=Reh˜1,h˜2,…,h˜M,Imh˜1,h˜2,…,h˜M∈RN×2M is the beam frequency matrix. Since ${\tilde{h}}_{m}$ is sparse, if the beam frequency matrix is regarded as a natural image, the compressed image restoration method can be used for beam space channel estimation. Therefore, a channel estimation network based on model-driven deep learning is developed.

## 3. Channel Estimation Network Based on Model-Driven Deep Learning

This article applies the Iterative Shrinkage Thresholding Algorithm (ISTA) [22] to the deep iterative network. The main update steps of the ISTA algorithm to restore the signal are as follows
(9)$$R^{\left(k\right)}={\hat{H}}^{\left(k-1\right)}-\rho {\bar{W}}^{⊤}\left(\bar{W}{\hat{H}}^{\left(k-1\right)}-Y\right)$$
(10)$${\hat{H}}^{\left(k\right)}=\underset{H}{\mathrm{argmin}}\frac{1}{2}{\left\|R^{\left(k\right)}-H\right\|}_{2}^{2}+\lambda \|\Psi H{\|}_{1}$$
wherein k represents the iteration index of ISTA, and ρ represents the iteration step size. Ψ is the transformation matrix that makes ΨH sparse in the transform domain. The two update steps (9) and (10) correspond to gradient descent and optimization procedures, respectively. Therefore, $R^{\left(k\right)}$ can be regarded as a noisy signal. In the broadband millimeter-wave system, to reduce the recovery error caused by manually setting parameters (such as λ, ρ and Ψ), this paper adopts the idea of learnable parameters, expands the iterative steps of the ISTA algorithm to ALISTA-Net, using the powerful learning power of neural networks to obtain an optimal set of parameters, and an adaptive selection threshold method based on ST-Net network is proposed in the beam domain denoising stage to reduce the loss-of-sparse recovery. The network consists of cascaded K layers with the same structure, which strictly corresponds to the K sub-iterative processes in the ISTA algorithm. The network structure is shown in Figure 2.

Taking the received signal Y and ${\hat{H}}^{{}^{\prime }(0)}$ as the input of the ALISTA-Net network, the estimation of the channel matrix ${\hat{H}}^{{}^{\prime }(K)}$ is used as the final output of the network. Using learnable parameters, learnable sparse matrices, and introducing an attention-based shrinkage threshold network (ST-Net) to replace manually set hyperparameters (ρ, λ, Ψ and thresholds), this work aims to select the best parameters during network training to compensate for recovery loss caused by improper manual settings. In this stage, the operation to expand to ALISTA-Net mainly includes the following processes:

1.Gradient descent process: $R^{\left(k\right)}$ can be rewritten as




(11)
$$R^{\left(k\right)}={\hat{H}}^{\left(k-1\right)}-\rho^{\left(k\right)}{\bar{W}}^{⊤}\left(\bar{W}{\hat{H}}^{{}^{\prime }(k-1)}-Y\right)$$



Among them, $\rho^{\left(k\right)}$ are the parameters that can be learned from the data, rather than manually setting each layer to share.

2.(Inverse) sparse transformation and residual learning process: The specific method of using sparse transformation is as follows



(12)
$$F^{\left(k\right)}\left(R^{\left(k\right)}\right)=B^{\left(k\right)}\cdot \mathrm{ReLU}\left(A^{\left(k\right)}\cdot {\left(R^{\left(k\right)}\right)}^{⊤}\right)$$



Among them, A(k)∈ℝω1×2M and B(k)∈ℝω2×ω1 are two learnable matrices, F(k)R(k)−F(k)(H)22≈α(k)R(k)−H22 as proved in [22], so the optimization process in Formula (10) can be rewritten as
(13)H^(k)=argminH12F(k)R(k)−F(k)(H)22+β(k)F(k)(H)1

$\beta^{\left(k\right)}=\lambda^{\left(k\right)}\times \alpha^{\left(k\right)}$, the optimization problem in Equation (13). can be viewed as a noise reduction problem, which is solved by the soft threshold operation in the frequency domain, i.e., $soft(x,\theta )=sign(x)(\max(0,\left|x\right|-\theta ))$, and sign(⋅) is a sign function. ${\tilde{F}}^{\left(k\right)}$ represents the inverse transformation that converts the signal from the sparse transform domain to the beam domain, $F^{\left(k\right)}$ and ${\tilde{F}}^{\left(k\right)}$ the network structure of and is symmetrical, consisting of two fully connected layers, but with different weights. So, the solution of Equation (13) is as follows
(14)$${\hat{H}}^{\left(k\right)}={\tilde{F}}^{\left(k\right)}\left(\mathrm{soft}\left(F^{\left(k\right)}\left(R^{\left(k\right)}\right),\theta^{\left(k\right)}\right)\right)$$

$\theta^{\left(k\right)}$ represents the k-th threshold of the contraction operation, which is also a learnable parameter. To speed up the convergence speed of the network, residual learning is introduced into ALISTA-Net, which is the part connected by the red line in Figure 2, so Equation (14) is rewritten as
(15)$${\hat{H}}^{\left(k\right)}=R^{\left(k\right)}+{\tilde{F}}^{\left(k\right)}\left(\mathrm{soft}\left(F^{\left(k\right)}\left(R^{\left(k\right)}\right),\theta^{\left(k\right)}\right)\right)$$

Modules $R^{{}^{\prime }(k)}$ and $R^{\left(k\right)}$ modules are similar, but the input is changed, as follows
(16)$$R^{{}^{\prime }(k)}={\hat{H}}^{\left(k\right)}-\rho^{{}^{\prime }(k)}{\bar{W}}^{⊤}\left(\bar{W}{\hat{H}}^{\left(k\right)}-Y\right)$$

To transform and denoise in the beam domain, it is first necessary for $R^{{}^{\prime }(k)}$ to perform a transpose operation. In the denoising stage, inspired by the deep residual shrinkage network, a shrinkage threshold network (ST-Net) based on the attention mechanism is proposed, which will be introduced later. Finally, the inverse transposition operation is performed, so the output can be expressed by the following formula
(17)$${\hat{H}}^{{}^{\prime }(k)}=R^{{}^{\prime }(k)}+T^{\prime }\left({\tilde{F}}^{\left(k\right)}\left(\mathrm{ST}\text{-}\mathrm{Net}\left(F^{{}^{\prime }(k)}\left(T\left(R^{{}^{\prime }(k)}\right)\right),\theta^{{}^{\prime }(k)}\right)\right)\right)$$

3.Shrink Threshold Network (ST-Net) Based on Attention Mechanism: The choice of threshold in Equation (17) has a great influence on the estimation accuracy of the ALISTA-Net model. Since the noise amount of each input is different, ST-Net is introduced in the beam domain denoising stage, inspired by the deep residual shrinkage network DRSN-CW structure in the literature [23]. The basic module is shown in Figure 3. The structure of the proposed shrinkage threshold network based on the attention mechanism is shown in Figure 4. First, the threshold in the soft threshold function is automatically learned and set by the network, which reduces the loss of accuracy caused by the inaccurate manual setting of the threshold. Secondly, the threshold value in the soft threshold function of the network is a positive number within the appropriate value range, to avoid the output situation of all zeros. At the same time, each sample has its own unique set of thresholds, making the model more applicable to situations with different noise contents.

The network structure draws on the idea of the attention mechanism. First, by taking the absolute value of the input, the absolute value of the sparse transformation feature map is simplified into a 16-dimensional vector by using Global Average Pooling (GAP). To make the intermediate output value of each layer the value more stable, and reduce the risk of model overfitting, the obtained features are input into two fully connected networks $N_{1}$ and $N_{2}$ with batch normalization (BN), and these two networks have 4 and 16 neurons, respectively, using Sigmoid and ReLU as activation functions to generate scaling vectors. The iteration threshold can be obtained by multiplying the dimensionality reduction vector and the scaling vector. The obtained threshold is a 16-dimensional vector, which changes in different iteration stages, and a set of thresholds for each channel can be adaptively obtained according to the characteristics of the input feature map. Finally, the contraction operation is performed according to the contraction function in the formula.

4.Training parameters: The trainable parameter set in ALISTA-Net is used Θ. According to the above description, Θ can be expressed as follows



$$\Theta =\left\{\rho^{\left(k\right)},\rho^{{}^{\prime }(k)},\theta^{\left(k\right)},F^{\left(k\right)}(\cdot )\right.,{\left.{\tilde{F}}^{\left(k\right)}(\cdot ),F^{{}^{\prime }(k)}(\cdot ),{\tilde{F}}^{{}^{\prime }(k)}(\cdot ),N_{1,k},N_{2,k}\right\}}_{k=1}^{K}$$



Among them, K is the total number of iterations of ALISTA-Net, and all parameters will be learned as parameters of the neural network.

5.Loss function: Since ALISTA-Net contains K stages, the loss function of the training process designed in this paper is as follows



(18)
$$\begin{array}{ll} L(\Theta ) & =L_{mse}+\mu L_{\mathrm{iteration}\;}+\zeta L_{\mathrm{symmetry}\;}\\ & ={\left\|{\hat{H}}_{K}-H\right\|}_{2}^{2}+\mu \sum_{k=1}^{K}{\left\|{\hat{H}}_{K}-H\right\|}_{2}^{2}+\zeta \sum_{k=1}^{K}{\left\|\tilde{F}\left(F\left({\hat{H}}_{K}\right)-H\right)\right\|}_{2}^{2} \end{array}$$



$L_{mse}$ is a measure of the loss between the output and the true value, and is a key component of the loss function. $L_{iteration}$ is the sum of the errors between the true value and the iterated value of ALISTA-Net in each stage. This helps to produce estimates that are close to the true value at each stage. $L_{\mathrm{symmetry}\;}$ is to establish $F\cdot \tilde{F}=I$ as much as possible.

## 4. Simulation Verification and Result Analysis

### 4.1. Simulation Data and Parameter Settings

To complete the training and testing of the ALISTA-Net network model, the specific parameter settings of the system model channel in this paper are shown in Table 1. the parameters of the training plan are set as shown in Table 2.

### 4.2. Simulation Result Analysis

In this paper, the normalized mean squared error (NMSE) is used as the difference between the channel matrix generated by the estimation estimate and the real channel matrix, which is defined as
(19)$$NMSE=E\left\{{\left\|H^{{}^{\prime }(K)}-H\right\|}_{2}^{2}/\|H{\|}_{2}^{2}\right\}$$
wherein the smaller the value of NMSE, the higher the accuracy of channel estimation by the model.

Ablation experiment
To compare the NMSE performance of the expanded iterative network (LISTA-Net) and the expanded iterative network (ALISTA-Net) using ST-Net, this paper designs an ablation simulation experiment for analysis. The simulation results are shown in Figure 5 and Figure 6.

As shown in Figure 5, the abscissa is the signal-to-noise ratio, and the ordinate is the normalized mean square error. As the SNR increases, the NMSE decreases gradually and the estimation accuracy increases. This is because as the signal-to-noise ratio increases, the amount of noise decreases gradually, and the impact on channel estimation gradually decreases, so the estimation accuracy gradually increases. Under five different signal-to-noise ratios, the NMSE of ALISTA-Net is lower than that of LISTA-Net, and the estimation accuracy is better than that of LISTA-Net. When SNR = −10, compared with LISTA-Net, ALISTA-Net has achieved 1.06 dB accuracy improvement. This is due to the proposed ST-Net adopting a strategy of providing different thresholds for each channel according to different noise characteristics instead of sharing thresholds for all channels.

In Figure 6, the abscissa is the number of layers deployed in the network, and the ordinate is the normalized mean square error. As the number of layers increases, the NMSE gradually decreases, and the convergence speed of ALISTA-Net is fast. Specifically, the ALISTA-Net method proposed in this paper can converge within six layers.

Table 3 lists the time it takes for the two methods to complete channel estimation in the online phase. The result obtained is the online test time per 10 epochs. The running speed can reflect the practicability of the algorithm in the time environment.

Since ALISTA-Net introduces the ST-Net module, the running time of ALISTA-Net is slightly higher than that of LISTA-Net, but the final time consumption is acceptable in the transmission of real-time communication systems.

Performance analysis
As shown in Figure 7, the ALISTA-Net algorithm proposed in this paper can not only obtain higher accuracy but also have a faster convergence speed than the current classic broadband system channel estimation method. Specifically, the proposed algorithm can converge within six layers; thus, the online stage can be deployed with a small number of layers, speeding up the training rate.

As shown in Figure 8, the ALISTA-Net method proposed in this paper achieves NMSE lower than that of the existing classic wideband system channel estimation algorithm in the range of 0–20 dB SNR. When SNR = 0, compared with the algorithm based on compressed sensing, ALISTA-Net obtains the largest gain and achieves NMSE of −16.24 dB, which verifies the effectiveness of the proposed scheme. Compared with deep learning-based ISTA-Net^+^ [24], a lower NMSE value is also achieved, which indicates that the proposed algorithm based on model-driven deep learning has been optimized for millimeter-wave massive MIMO broadband system channel estimation.

Complexity analysis
As shown in Table 4, the complexity of the proposed ALISTA-Net is compared with other channel estimation methods, and ALISTA-Net achieves a lower computational complexity than LDGEC and ISTA-Net^+^. This is because the computational complexity of ALISTA-Net is mainly determined by matrix multiplication, and the computational complexity of LDGEC is mainly determined by matrix inversion. The computational complexity of ISTA-Net^+^ is mainly determined by the convolution operation, where K is the filter size, $C_{in}$ and $C_{\mathrm{out}}$ are the number of input channels and output channels of the convolution, respectively (in [21], $C_{in}=C_{\mathrm{out}}=32$). Although the number of trainable parameters of ALISTA-Net is higher than that of LISTA-Net and ISTA-Net^+^, it is known that both ALISTA-Net achieves better estimation accuracy than LISTA-Net and ISTA-Net^+^ based on the results of ablation experiments and performance tests.

## 5. Conclusions

Aiming at the problem of low estimation accuracy of existing channel estimation algorithms in millimeter-wave massive MIMO broadband systems with a low signal-to-noise ratio, this paper proposes a channel estimation algorithm based on model-driven deep learning considering the “beam squint” effect. Ablation experiments show that, compared with LISTA-Net without ST-Net, ALISTA-Net achieves a higher estimation accuracy and convergence speed in low SNR areas. Performance experiments show that compared with existing wideband system algorithms, ALISTA-Net can still achieve higher estimation accuracy in areas with low SNR. Therefore, the ALISTA-Net proposed in this paper is suitable for channel estimation in millimeter-wave massive MIMO broadband systems and has faster convergence speed and higher channel estimation accuracy.

## Figures and Tables

**Figure 1 sensors-23-02638-f001:**
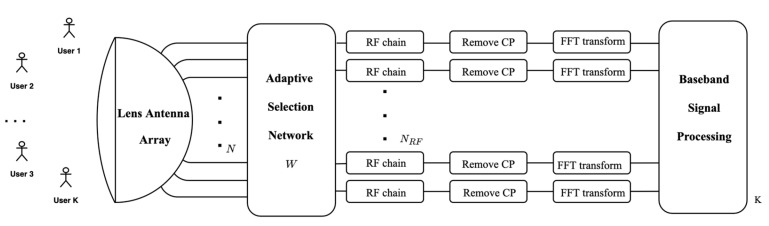
Broadband millimeter-wave MIMO-OFDM system structure based on lens antenna array.

**Figure 2 sensors-23-02638-f002:**
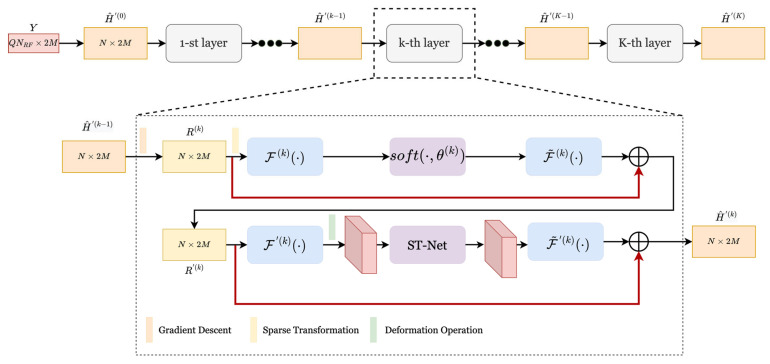
ALISTA-Net network structure diagram.

**Figure 3 sensors-23-02638-f003:**
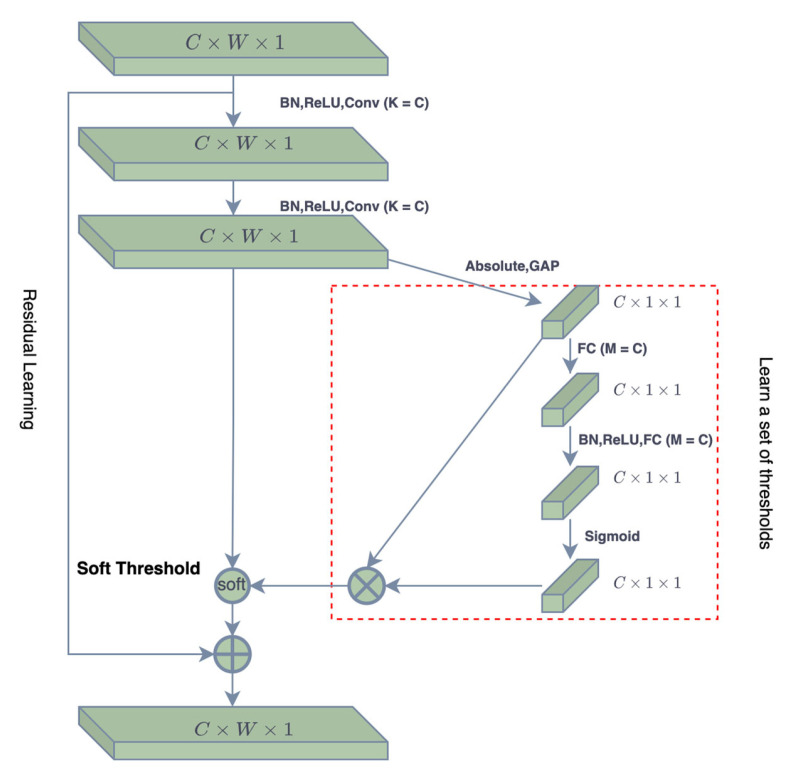
The basic module of deep residual shrinkage network.

**Figure 4 sensors-23-02638-f004:**
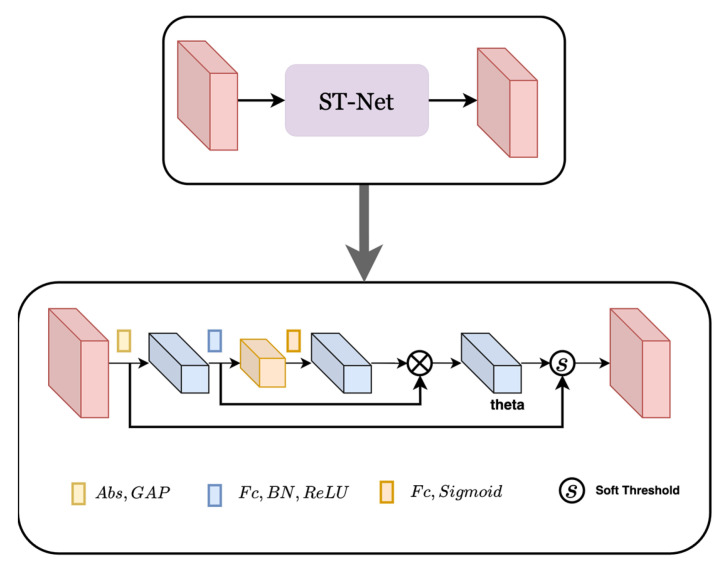
ST-Net network structure diagram.

**Figure 5 sensors-23-02638-f005:**
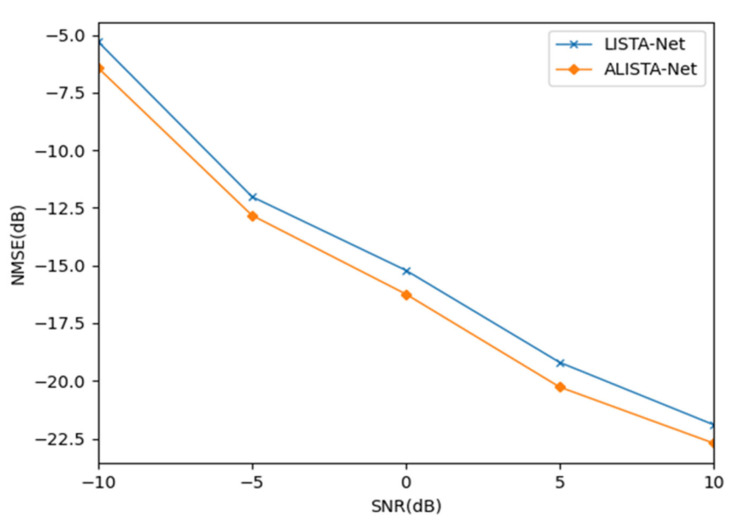
Performance Comparison of ALISTA Net and LISTA Net under Different SNRs.

**Figure 6 sensors-23-02638-f006:**
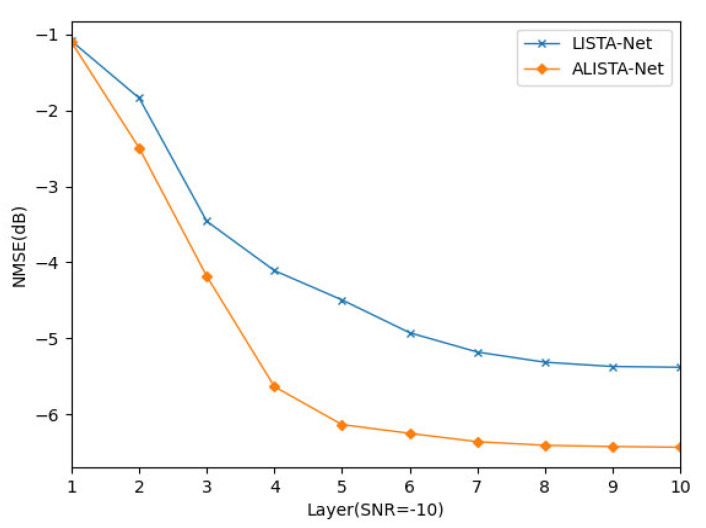
Convergence comparison between ALISTA Net and LISTA Net when SNR = −10.

**Figure 7 sensors-23-02638-f007:**
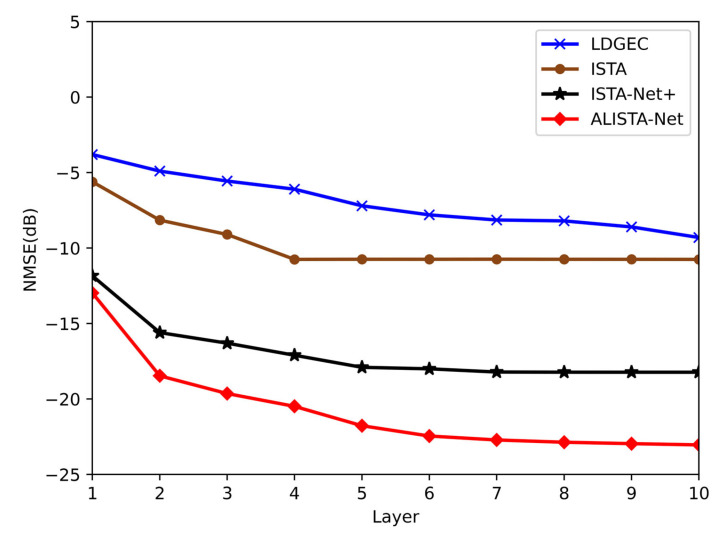
Convergence comparison between ALISTA Net and other channel estimation algorithms.

**Figure 8 sensors-23-02638-f008:**
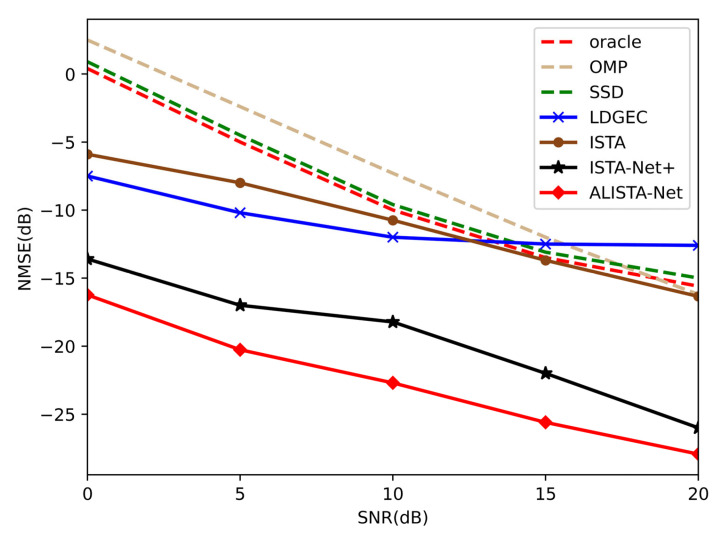
Comparison of NMSE performance between ALISTA Net and other channel estimation algorithms.

**Table 1 sensors-23-02638-t001:** Channel Parameter Setting of Millimeter-Wave Massive MIMO System.

Parameter	Value
Number of lens elements	32
Number of RF chains	8
Carrier frequency	28 GHz
Bandwidth	4 GHz
Number of subcarriers	32
Maximum delay	20ns
Physical direction of the path	$\theta_{l}∼\mu (-\pi /2,\pi /2)$
Delay of the path	$\tau_{l}∼\mu (0,20ns)$
Number of resolvable paths	L=3

**Table 2 sensors-23-02638-t002:** Training plan parameter settings.

Parameter	Value
Training set	10,000
Validation set	1280
Testing set	2560
Batch Size	64
Optimizer	Adam
Learning rate	0.0001
SNR	[−10,20]
Maximum training iterations	5000

**Table 3 sensors-23-02638-t003:** Running time.

Algorithm	Time/s
LISTA-Net	9.17
ALISTA-Net	9.25

**Table 4 sensors-23-02638-t004:** Complexity Analysis.

Method	Parameters	Computational Complexity
LISTA-Net	$1.65\times 10^{5}$	$O(QN_{RF}NM)$
SSD	0	$O(MN_{RF}QL^{2}\Omega^{2})$
OMP	0	$O(MN_{RF}QL^{3}\Omega^{3})$
LDGEC	$5.19\times 10^{5}$	$O(MN^{3})$
ISTA	2	$O(QN_{RF}NM)$
ISTA-Net^+^	$3.76\times 10^{4}$	$O(MNK^{2}C_{in}C_{out})$
ALISTA-Net	$1.68\times 10^{5}$	$O(QN_{RF}NM)$

## Data Availability

The processed data required to reproduce these findings cannot be shared as the data also form part of an ongoing study.

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
