# Peer review of "A Model-Driven Channel Estimation Method for Millimeter-Wave Massive MIMO Systems"

_sensors, 2023, doi:10.3390/s23052638_

Round 1

Reviewer 1 Report

This manuscript proposes a new channel estimation method based on the Adaptive Threshold Learning Iterative Shrinkage Threshold Network (ALISTA-Net) by taking into account the beam squint effect. The proposed estimation method can achieve better performance in terms of NMSE results. In addition, the topic is timely and interesting. However, the reviewer has these comments:

  1. 1-  In the introduction section, the authors should change or enhance the following sentence because the CSI is indispensable for any communication system and represent the difficulty of mmWave channel estimation in the presence of the Beam Squint effect, especially in mmWave MIMO OFDM systems

    " In massive MIMO, accurate channel state information (CSI) is required for both channel equalization and symbol detection at the receiving end. And the spatial high resolution can only be achieved if the propagation environment is accurately known, therefore, accurate channel estimation becomes a major factor limiting the performance of massive MIMO systems.”

  2. 2-  "However, the performance of the compressed sensing algorithm is greatly affected by the prior sparsity and the iteration step size, and the anti-noise ability of the algorithm is poor in the environment of low signal-to-noise ratio, so it is not suitable for the channel estimation of the millimeter-wave massive MIMO broadband system."

    To confirm this information, the authors may compare the NMSE results of the model-driven channel estimation method with the performance of the greedy algorithm such as orthogonal matching pursuit (OMP) and generalized orthogonal matching pursuit (gOMP) and represent the oracle NMSEs results as a lower bound in the comparison for illustrating the estimation accuracy.

  3. 3-  The main drawback of the model-driven channel estimation method is that there are no analytical solutions to find suitable parameters (?, ?, Ψ, and thresholds). Therefore, the authors should explain how to initialize these parameters.

  4. 4-  This model-driven channel estimation method can achieve a higher estimation accuracy when the BS has a higher number of lens elements (N = 256 element lens) ?

  5. 5-  For reference [19], the paper title is not correct.

Reviewer 2 Report

1.     Recently, a novel MIMO concept, named index modulation (spatial modulation), is able to achieve the higher capacity than conventional MIMO scheme [1-2]. In addition, the channel estimation for spatial modulation is also well investigated [3]. Therefore, to improve the quality of this paper, the concept of spatial modulation and its channel estimation are better to introduce in Introduction of this paper.

[1] A Survey on Spatial Modulation in Emerging Wireless Systems: Research Progresses and Applications, IEEE JSAC, 2019.

[2] Composite Multiple-Mode Orthogonal Frequency Division Multiplexing with Index Modulation, IEEE Transactions on Wireless Communications, 2022.

[3] Channel Estimation for Spatial Modulation, IEEE Transactions on Communications, 2014.

2.     How to optimally decide the threshold for the iterative algorithm?

3.     The complexity comparisons of the proposed methods and conventional methods should be included, which further reveals the advantages of the proposed method.

4.     The accuracy of channel estimation should be provided for the proposed and conventional methods.

5.     Please draw all curves smoother. In addition, please improve the resolution of all figures.

6.     Please correct some typo errors.

Reviewer 3 Report

The topic of this paper is interesting and timely but the reviewer has some concerns,

·         In the introduction you mentioned “giant connection”, do not use the expression “giant”

·         In the introduction is mentioned “Aiming at the above problems, this paper proposes a new channel estimation method based …”. What problems? Please be more specific and improve the contributions part of this work.

·         The SoA should be improved by adding more recent references, in the last 2-3 years some work on this topic have been published.

·         The math formulation is quite confused since the same formatting is used for matrices, vectors and scalars. The reviewer suggest to use the conventional notation: bold upper case for matrices, bold lower case for vectors and italic for scalars.

·         The complexity of the proposed solution is not analyzed, which is important for practical applications.

·         The English and the organization should be improved.

Round 2

Reviewer 1 Report

The authors have taken into consideration the remarks I have made. Therefore, I believe that the article can be accepted for publication.

Reviewer 2 Report

This paper is well revised. No further comments.

Reviewer 3 Report

The authors have satisfactory addressed the reviewer concerns. However the English can be further improved.